# Improved Through-Plane Thermal Conductivity of Poly(dimethylsiloxane)Composites through the Formation of 3D Filler Foam Using Freeze-Casting and Annealing Processes

**DOI:** 10.3390/nano13152154

**Published:** 2023-07-25

**Authors:** Jooyoung Lee, Wonyoung Yang, Geunhyeong Lee, Youngsung Cho, Jooheon Kim

**Affiliations:** 1School of Chemical Engineering and Materials Science, Chung-Ang University, Seoul 06974, Republic of Korea; jooyoung0426@naver.com (J.L.); yangwonyoung1007@gmail.com (W.Y.); rmsgud96@naver.com (G.L.); ysc7817@naver.com (Y.C.); 2Department of Advanced Materials Engineering, Chung-Ang University, Anseong 17546, Republic of Korea; 3Department of Intelligent Energy and Industry, Graduate School, Chung-Ang University, Seoul 06974, Republic of Korea

**Keywords:** composites, annealing, thermal properties, freeze drying

## Abstract

The configuration of a continuous and oriented thermal pathway is essential for efficient heat dissipation in the oriented direction. Three-dimensional (3D) conductive filler structures provide a suitable approach for constructing continuous thermal pathways in polymer-based composites. The aluminum nitride/reduced graphene oxide/poly(dimethylsiloxane) (AlN/rGO/PDMS) composite material is made with a 3D foam structure and focuses on reducing GO and forming foam via polyvinyl alcohol (PVA). We analyze the successful fabrication of hybrid fillers and composites using various methods. The fabricated composite with a 3D network filler foam achieves a through-plane thermal conductivity of 1.43 W/mK and achieves 752% higher thermal conductivity compared to pure PDMS, which is superior to composites without 3D foam. The continuous 3D filler structure via freeze-drying and annealing processes provides efficient thermal dissipation in the through-plane direction pathway, which is critical for enhancing thermal conductivity. Therefore, this work produces a polymer composite material with improved thermal conductivity through various processes.

## 1. Introduction

As technology advances, the integration of electronic materials also improves, resulting in the generation of a significant amount of heat. The efficient dissipation of heat is crucial for maintaining device performance by managing its temperature [1,2,3]. The heat sink, known for its high thermal conductivity, rapidly dissipates the generated heat into the surroundings. Between the heat sink and the heating element, which fulfills this role, there exists a Thermal Interface Material (TIM) layer that necessitates high thermal conductivity [4,5,6]. TIMs are commonly prepared using epoxy- or silicone-based polymers [7,8], but their intrinsic low thermal conductivity fails to meet the thermal requirements of electronic devices. To enhance thermal conductivity, the incorporation of highly thermally conductive fillers, such as aluminum nitride (AlN) [9,10,11,12] and boron nitride (BN) [13,14,15,16], into the polymer matrix is employed. Notably, graphene oxide (GO) is an excellent material for TIMs due to its high thermal conductivity, lightweight nature, flexibility, and good compatibility. The construction of a continuous thermal pathway using anisotropic filler orientation offers advantages in enhancing the thermal conductivity of polymer composites [17,18].

Several studies have focused on the formation of oriented heat transfer pathways in polymer-based composites to achieve efficient heat dissipation. For instance, Hu et al. utilized the hot press method to obtain oriented boron nitride (BN), resulting in a significant improvement in thermal conductivity along the orientation direction of the composite material [19]. Wang et al. fabricated an oriented hBN/PS composite material through latex mixing and compression molding. The in-plane orientation of hBN nanosheets in the composite material led to an excellent in-plane thermal conductivity of 8.0 W m^−1^ K^−1^ [20]. Chen et al. employed electrospinning to connect and vertically align boron nitride nanosheets, thereby enhancing the thermal conductivity of the polymer [20]. Chen et al. demonstrated that, by electrospinning, the boron nitride nanosheets were connected to each other and aligned vertically to improve the thermal conductivity of the polymer [21]. In previous studies, 3D filler networks were constructed in polymer composites via freeze drying [22,23], vacuum infiltration [24,25], and chemical vapor deposition (CVD) [26,27]. The fillers were oriented according to the pre-formed network foam, successfully fabricating a composite with improved thermal conductivity.

Here, we present a novel method for fabricating AlN/reduced graphene oxide (rGO) foams through freeze-casting and PVA annealing processes. In this method, the annealing of PVA creates a heat conduction path along the produced 3D network foam, playing a critical role in improving thermal properties. The foam, acting as a filler, plays a crucial role in enhancing thermal properties. During the annealing process, the reduction of GO to rGO occurs, further contributing to the enhancement of thermal properties.

## 2. Experimental Section

### 2.1. Materials and Methods

Aluminum nitride (AlN with 32.0% N min, density 3.3 g cm^−3^ and average diameter 4 μm) is obtained from Alfa Aesar. Graphene oxide power is purchased from Grapheneall (Siheung, Republic of Korea). PVA (MW 31,000~50,000, 98~99% hydrolyzed) is obtained from Sigma-Aldrich (St. Louise, MO, USA). Poly(dimethyl siloxane) (PDMS) and a curing agent (SYLGARD™ 184 silicon elastomer) are bought from Dow Silicones Corporation (Carrollton, KY, USA).

### 2.2. Preparation of AlN/GO/PVA Foam

GO (250 mg) and PVA (31.25 mg) are currently being sonicated for 30 min to disperse them in deionized water. The mass ratio between GO and PVA has been set based on the minimum amount of PVA required to maintain the 3D filler structure. Following that, AlN (1.25 g) will be added to the GO/PVA solution, and the mixture will be stirred at room temperature for 12 h to create an AlN/GO/PVA solution. The well-dispersed solution then is poured into a mold and subjected to a 48 h lyophilization process. Consequently, an AlN/GO/PVA foam with a three-dimensional network is fabricated.

### 2.3. Preparation of AlN/rGO/PDMS

The AlN/GO/PVA foam is produced through freeze casting and undergoes annealing to produce the AlN/rGO foam. PVA vaporizes at temperatures above 200 °C, while GO is reduced at 330 °C. Therefore, the annealing process is carried out at 450 °C for 2 h [1]. This process involves raising the temperature of the GO/PVA foam, resulting in a structure composed solely of AlN/rGO fillers. Subsequently, the AlN/rGO/PDMS composite is fabricated by immersing the foam into a flexible and interfacially compatible PDMS and curing agent (in a 10:1 weight ratio).

### 2.4. Characterization

The synthesis of hybrid filler and 3D foam is being studied using Fourier transform infrared spectroscopy (FT-IR). Specifically, the Perkin–Elmer Spectrum One instrument is being used in ATR mode for analysis. Thermogravimetric analysis (TGA) is being investigated using the TGA-2050 instrument, with a temperature range of 25–800 °C and a heating rate of 20 °C min^−1^. To analyze the formation of the 3D filler structure and obtain cross-sectional images of the composites, field emission transmission electron microscopy (FE-SEM, Carl Zeiss, Oberkochen,, Germany) is being conducted at 5 kV after platinum surface coating. Energy-dispersive X-ray spectroscopy (EDS, ThermoNORAN System 7, Waltham, MA USA) is being used at 30 kV to verify the even distribution of AlN in the composite. The laser flash method (LFA, NanoFlash 467, Netzsch Instruments Co., Burlington, MA, USA) is being applied to measure the thermal diffusivity of the sample at room temperature, using a sample diameter of 25 mm. Universal Testing Machine (UTM, UTM-301, R&B Corp., San Jose, CA, USA) is being employed to evaluate the mechanical properties of the fabricated composites. Two methods are being used to obtain the porosity of the 3D foam: Brunauer–Emmett–Teller (BET) nitrogen adsorption and mercury intrusion porosimetry (MIP). These methods provide information about the porosity characteristics of the foam structure.

## 3. Results and Discussion

### 3.1. Schematic Illustration of the Composite Fabrication

To enhance the through-plane thermal conductivity in the composite material, we prepare a 3D network foam consisting of AlN, GO, and PVA. AlN/GO/PVA solution is mixed by adding AlN to the sonicated GO/PVA solution, and then manufacture the AlN/GO/PVA foam through freeze-drying. Afterwards, PVA is removed through annealing and infiltrate PDMS to finally produce AlN/rGO/PMDS composites. Figure 1 shows the manufacturing process of the composite.

### 3.2. Thermal Properties and Morphologies of PVA Composites

The FT-IR spectra of raw AlN, raw GO, PVA, and AlN/GO/PVA film are shown in Figure 2a. The hydroxyl group’s OH stretch is responsible for the peak observed at 3360 cm^−1^ [28]. The asymmetric and symmetrical stretching of CH_2_ in GO is represented by the peak at 2802 cm^−1^, while the 1625 cm^−1^ peak corresponds to the C=C stretching of graphite [29]. Furthermore, peaks at 1731 cm^−1^ indicate the C=O vibration of the carboxyl group [30], the 1224 cm^−1^ peak signifies the C-OH bond of the alcohol group [31], and the 1080 cm^−1^ peak corresponds to the C−O stretch of C−O−C [32]. In the FT-IR spectrum of AlN/GO/PVA, all these constituent material peaks are observed, confirming the successful fabrication of the composite. Figure 2b shows the TGA curve of raw AlN, raw GO, PVA, and AlN/GO/PVA composite film. The TGA curve of pure PVA exhibits a weight loss area similar to that of previous studies [33]. A mass loss between 200 °C and 300 °C due to the loss of water molecules from the polymer matrix. Between 300 °C and 500 °C, further mass loss occurs due to the decomposition and carbonization of the polymer. The weight loss before 100 °C in GO is attributed to the loss of physiosorbed water molecules [34]. The weight loss in the range of 200 °C to 250 °C and 250 °C to 600 °C is due to the less stable pyrolysis of the oxygenated functional groups [35]. The TGA graph of AlN/GO/PVA composites film confirms the mass reductions shown in raw GO and PVA. At approximately 200 °C, the mass of GO decreases, and at approximately 400 °C, the mass of PVA decreases, showing a decreasing trend in the graph.

### 3.3. Morphology of the AlN/rGO Foam

Figure 3a,b are FE-SEM images of raw material PVA (at a 744 magnification) and AlN/GO/PVA (at a 9.12 × 10^3^ magnification) prepared by the freeze-drying. Figure 3b confirms that AlN and GO are well dispersed. Figure 3c is an FE-SEM image of AlN/rGO foam (at a 6.74 × 10^3^ magnification), showing that PVA was removed through annealing and the filler foam was maintained. Figure 3d presents the FT-IR spectra of AlN/GO/PVA and AlN/rGO foam. In the case of AlN/rGO foam, it is observed that the -OH peak (3360 cm^−1^) is absent, indicating successful reduction of GO during the annealing process, thus validating the experiment’s success. To confirm the presence of AlN and rGO fillers in the foam, TGA analysis is conducted. The TGA graph in Figure 3e illustrates the disparity in mass loss between AlN/GO/PVA and AlN/rGO in the temperature range of 200 °C to 400 °C, attributed to the reduction of GO and removal of PVA through the annealing process. To analyze the 3D porous structure of AlN/rGO foam, N_2_ adsorption–desorption isotherms curve and pore size distribution of AlN/rGO and AlN/rGO foam via freeze-drying are shown in Figure 4a,b. BET is a method that can be analyzed for open pores by measuring the pores size distribution by adsorbing nitrogen gas to the sample [36]. The more the sample adsorbs nitrogen the higher AlN/rGO foam via freeze-drying performs strong nitrogen adsorption in low relatively low pressure (P/P_0_ < 0.001) and high pressure (P/P_0_ > 0.8), which indicates the 3D filler foam has micropores [37,38,39]. For AlN/rGO filler showing low adsorption at low relative pressure means that the filler rarely has a microporous structure. In a pore size distribution graph, AlN/rGO filler has nanosized pores which is consistent with nitrogen adsorption–desorption isotherm analysis. AlN/rGO foam, on the other hand, has both nano-sized and micro-sized pores due to the freeze-drying and annealing process. Consequently, these morphologies and pore size distribution analyses verify that the 3D porous structure of the filler foam is successfully fabricated.

### 3.4. Morphology of the AlN/rGO/PDMS Composite

Figure 5a,b shows FE-SEM images of AlN/rGO/PDMS. As shown in Figure 5a (at a 2.08 × 10^3^ magnification), the AlN/rGO/PDMS composite fabricated by random dispersion is generated indiscriminately without forming a network. Furthermore, the aggregation of the hybrid fillers occurred, which is the main factor hindering the improvement of thermal conductivity. On the other hand, in Figure 5b (at a 2.51 × 10^3^ magnification), filler foam formed through freeze-drying and annealing processes can be found in composite materials and improves thermal conductivity by forming a three-dimensional heat transfer path. To confirm that the filler network was maintained, the EDS mapping image of AlN/rGO/PDMS was analyzed (Figure 5c). The even distribution of Si elemental confirms the successful infiltration of the PDMS matrix via vacuum-assisted infiltration. The presence of C, O, and Al elements, which are the main components of the AlN/rGO foam, indicates a similar pattern, verifying that the 3D structure through the freeze-drying and annealing process is well-executed. These results demonstrate the successful and uniform construction of the 3D foam structure within the matrix, allowing for the formation of an efficient thermal pathway.

### 3.5. Thermal and Mechanical Properties of the AlN/rGO/PDMS Composite

The formula for calculating thermal conductivity is λ = α · Cp · ρ. λ is the thermal conductivity (W m^−1^ K^−1^), α is the thermal diffusivity (m^2^/s), Cp is the specific heat at constant pressure (J/Kg·K), ρ is the density (Kg/m^3^). The thermal conductivity according to the filler content of the PDMS-based composite material is shown in Figure 6a. As the filler loading in the composites increases, the thermal conductivity of the composites also gradually improves. The pure PDMS thermal conductivity is only (0.19) W m^−1^ K^−1^, but the AlN/rGO/PMDS composite (filler content 50 wt.%) has improved through-plane thermal conductivity to achieve 1.43 W m^−1^ K^−1^. The AlN/rGO/PDMS composite shows improved through-plane thermal conductivity with 50 wt.% of hybrid filler loading (1.43 W m^−1^ K^−1^). The fabricated composite improved by 752% compared to pure PDMS. Figure 6b shows the thermal conductivity enhancement (TCE) of the fabricated composites with various processes with the 50 wt.% filler contents. The calculation of TCE is given in Formula (1) as follows:(1)$$\mathrm{TCE}=\frac{K_{f}-K_{m}}{K_{m}}$$

The thermal conductivity of the composite and matrix are expressed as *K_f_* and *K_m_*, respectively. The composite prepared with freeze drying method depicts improved thermal conductivity by 137% compared to the randomly dispersed composite due to the continuous 3D heat pathway. Therefore, the heat dissipation along the oriented thermal pathway of the resulting AlN/rGO/PDMS composite after forming the 3D network foam using the freeze-drying and annealing processes are superior to other methods. To verify the superior thermal properties of AlN/rGO/PDMS, we compare to other previous studies using various fabrication methods, as shown in Figure 6c [40,41,42,43,44,45,46]. The mechanical properties of the composite are analyzed using UTM. The compatibility of the matrix and the filler and the aggregation of the hybrid filler affect the mechanical properties of the composites critically. The tensile strength and tensile strain of the AlN/rGO/PDMS composite filler according to the filler content differ according to the difference in interfacial compatibility and dispersibility, and the comparison values are shown in Figure 6d. As the filler content increases, it forms a 3D network, connecting in a through-plane direction rather than an in-plane direction. Therefore, a higher filler content increases tensile strength but decreases tensile strain. Figure 6e is the tensile strength–strain graph according to the manufacturing process of the AlN/rGO/PDMS composite material. The freeze-dried AlN/rGO/PDMS composite (1.7 MPa) shows higher mechanical properties including tensile strength and tensile strain than randomly dispersed composites (1.35 MPa) due to their network structure. Therefore, there is considerable potential for the TIM development approach.

## 4. Conclusions

To enhance the thermal management performance of polymer-based composites, researchers are currently focusing on establishing effective heat transfer pathways. Oriented filler networks play a vital role as heat transfer pathways in polymer composites. In this study, we fabricate the AlN/rGO/PDMS composite to achieve a successful three-dimensional (3D) network. It is accomplished by creating a foam structure through freeze-drying and subsequent annealing processes. The fabricated composite, which features a 50 wt.% filler loading and a 3D foam structure, exhibits a remarkable 752% improvement in thermal conductivity compared to pure PDMS. Notably, thermal conductivity enhancement of AlN/rGO/PDMS via freeze-drying surpasses that of randomly dispersed composite materials. The main factor of significant increase in thermal conductivity is the well-established thermal conductivity pathway formed by the 3D network of the filler, which contributes to the formation of the newly arranged foam structure. The findings of our study present an effective and straightforward approach to greatly enhance the thermal conductivity of materials used in thermal management applications. By utilizing a relatively simple fabrication process, the composite material achieves superior thermal performance, highlighting its potential for practical implementation in various thermal management systems.

## Figures and Tables

**Figure 1 nanomaterials-13-02154-f001:**
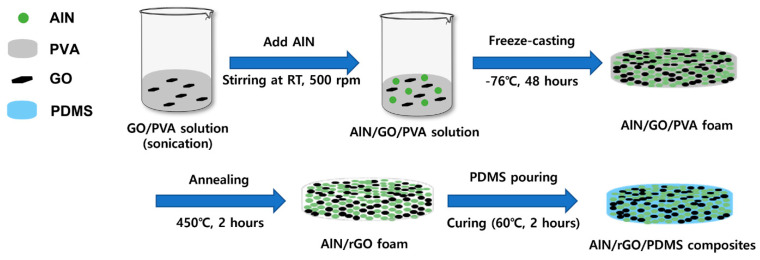
A schematic diagram of AlN/rGO/PDMS composite fabrication process.

**Figure 2 nanomaterials-13-02154-f002:**
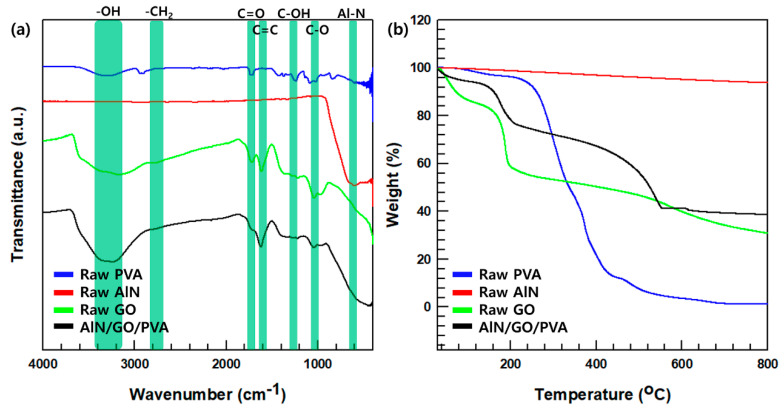
Analysis for raw PVA, raw AlN, raw GO and AlN/GO/PVA. (**a**) FT-IR spectra. (**b**) TGA curves of materials.

**Figure 3 nanomaterials-13-02154-f003:**
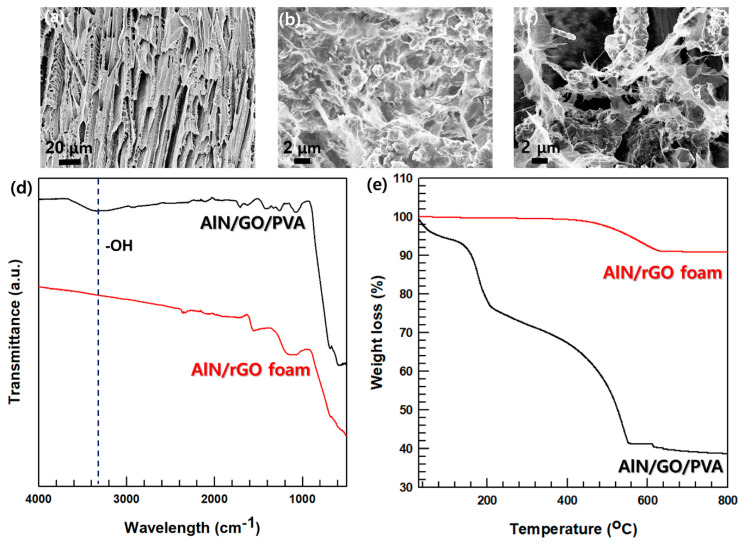
FE-SEM Images of composites; (**a**) PVA foam, (**b**) AlN/GO/PVA, (**c**) AlN/rGO foam. Analysis of AlN/GO/PVA and AlN/rGO foam; (**d**) FT-IR, (**e**) TGA.

**Figure 4 nanomaterials-13-02154-f004:**
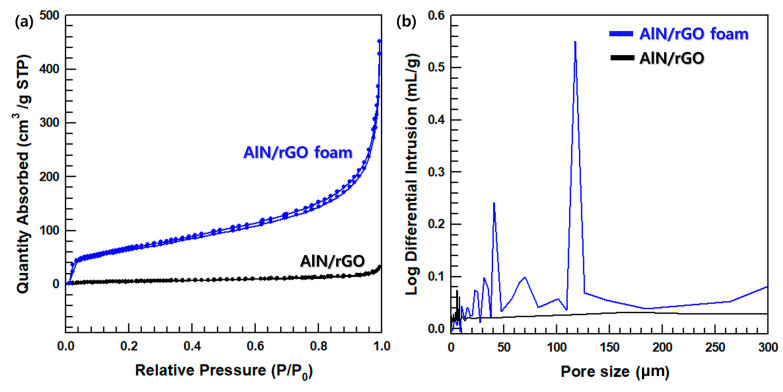
(**a**) Nitrogen adsorption–desorption isotherms curve, and (**b**) pore size distribution of AlN/rGO foam and AlN/rGO.

**Figure 5 nanomaterials-13-02154-f005:**
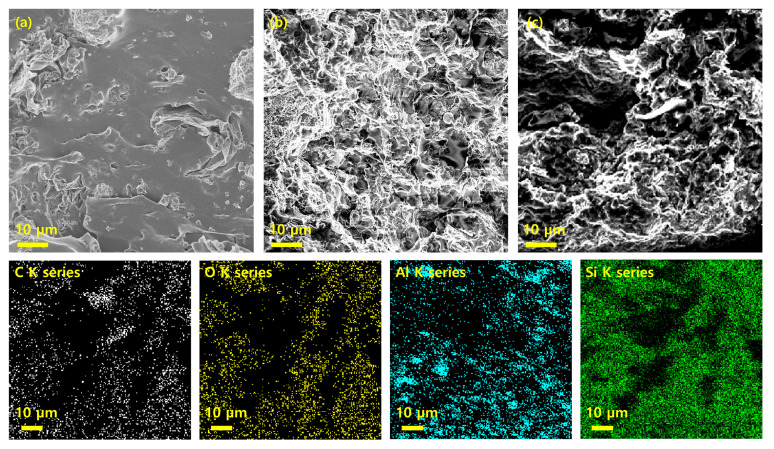
Cross-sectional FE-SEM image of AlN/rGO/PDMS composite (filler content 50 wt.%); (**a**) random dispersed (**b**) freeze-drying. (**c**) Cross-sectional FE-SEM image of AlN/rGO/PDMS via freeze drying and EDS mapping of C, O, Al, and Si elements.

**Figure 6 nanomaterials-13-02154-f006:**
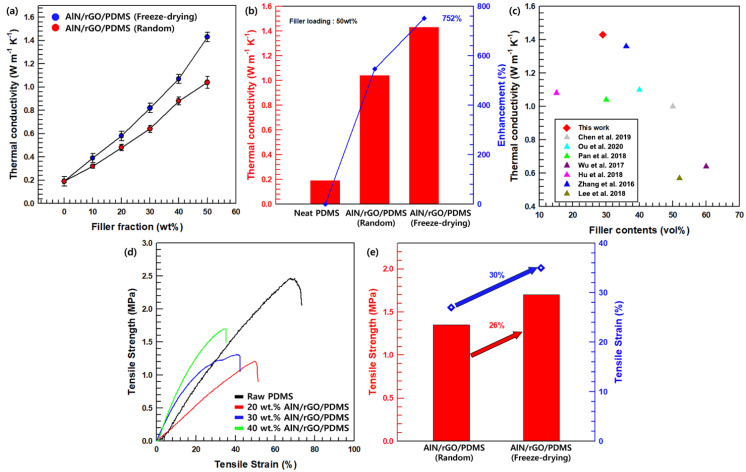
Through-plane thermal conductivity and TCE; (**a**) different filler loadings (**b**) different manufacturing methods (**c**) comparison of thermal conductivity of AlN/rGO/PMS composites with other composites [40,41,42,43,44,45,46]. Tensile stress-tensile strain graph; (**d**) different filler loadings; and (**e**) different manufacturing methods.

## Data Availability

All data generated or analyzed during this study are included in this published article.

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
