# Peer review of "Improved Through-Plane Thermal Conductivity of Poly(dimethylsiloxane)Composites through the Formation of 3D Filler Foam Using Freeze-Casting and Annealing Processes"

_nanomaterials, 2023, doi:10.3390/nano13152154_

Round 1
Reviewer 1 Report
Interesting work but the following comments to improve the manuscript
1. Provide SEM images with their scale and magnification details
2. Suggesting authors to provide cross-sectional SEM images with elemental mapping
3. Suggesting authors provide relevant data for porosity (if necessary BET data)
4. Improve the Conclusion section
5. Figures quality should be improved.
Typo errors should be checked and verified.
Reviewer 2 Report
How to design materials to achieve continuous directional thermal channels and achieve effective heat dissipation is a research hotspot. In this paper, authors prepare the three-dimensional (3D) conductive filler structure which can provide a suitable method for constructing continuous thermal channels in polymer based composites. Here they provide a method for Preparation of AlN/GO/PVA foam, and analyze thermal and mechanical properties of the AlN/rGO/PDMS composites. They obtain some interesting results. I would like to recommend it for publication in Nanomaterials after modifying some minor questions:
(1) The format of the references needs to be carefully checked.
(2) I suggest using the present tense instead of the past tense to describe their work in the paper..
Reviewer 3 Report
1. Please give the purity of samples.
2. Why was the mass fraction for GO (250 mg) and PVA (31.25 mg) selected? Different mass fraction has different PPI foam, right? If yes, then what foam did authors hope to get? Please give the detailed explanation.
3. In figure 1. The schematic diagram seems to be a little simple, please add the temperature condition and other information, thus, the readers are easy to understand the process.
4. Why did the wavenumber Vs Transmittance be used? Please give the explanation.
5. What is the through-plane thermal conductivity?
6. Page 5, ‘The fabricated composite was improved by 752% compared to pure PDMS.’, why did authors have a comparison with PDMS?
7. How to get the figure 5? By experiment of this work, or other literatures? It is unclear for the expression of this section.
8. Why was the equation (1) used to obtained TCE? Or why was it suitable to get the TCE? Please give the reason.
Please check the expression and grammar, especially grammar.
Author Response
Please see the attachment. We also fixed the grammatical and typo errors.

Round 2
Reviewer 3 Report
Accept
Fine